# Hallucination Detox:
# Sensitivity Dropout (SenD) for Large Language Model Training

## Abstract

As large language models (LLMs) are increasingly deployed across various industries, concerns regarding their reliability, particularly due to hallucinations—outputs that are factually inaccurate or irrelevant to user input—have grown. Our research investigates the relationship between the training process and the emergence of hallucinations to address a key gap in existing research that focuses primarily on post hoc detection and mitigation strategies. Using models from the Pythia suite (70M–12B parameters) and several hallucination detection metrics, we analyze hallucination trends throughout training and explore LLM internal dynamics. We introduce **Sensitivity Dropout (SenD)**, a novel training protocol designed to mitigate hallucinations by reducing variance during training. SenD achieves this by deterministically dropping embedding indices with significant variability, referred to as Sensitive Embedding Indices. In addition, we develop an unsupervised hallucination detection metric, Efficient EigenScore (EES), which approximates the traditional EigenScore in 2x speed. This efficient metric is integrated into our protocol, allowing SenD to be both computationally scalable and effective at reducing hallucinations. Our empirical evaluation demonstrates that our approach improves LLM reliability at test time by up to 40% compared to normal training while also providing an efficient method to improve factual accuracy when adapting LLMs to Wikipedia, Medical, and LegalBench domains.

## 1 Introduction

### 1.1 Motivation

As Large Language Models (LLMs) become more sophisticated and widespread across industries, concerns about their reliability and safety have grown due to misuse and user errors. One of these concerning areas discovered by the scientific community is the phenomenon of hallucinations - LLMs producing content that may not align with real-world facts, the user's input, or training data it has seen in the past (Huang et al., 2023a). In our research, we target a specific field of hallucinations called confabulations which occur when the LLM generates different responses given the same or similar inputs. This can be harmful when the generations alter between correct and factually incorrect responses.

Previous research has largely focused on identifying and addressing hallucinations in large language models (LLMs), but the impact of the training process on hallucinations remains under-explored (Huang et al., 2023a; Rawte et al., 2023; Ye et al., 2023; Hong et al., 2024; Xu et al., 2024; Chen et al., 2024; Li et al., 2024; Gao et al., 2024b). This paper addresses this gap by investigating how the iterative learning process in LLMs leads to significant variance in hallucination behavior during training. This variability indicates that the model's factual confidence fluctuates, making it challenging to pinpoint a checkpoint at which the model has confidently learned facts.

As LLMs are deployed in high-risk industries, ensuring their reliability is crucial for user safety. However, this is not always achieved, leading to serious consequences, such as an Air Canada lawsuit over an LLM-generated incorrect policy (Garcia, 2024). Addressing such issues requires a

deeper understanding of how hallucinations arise during training, enabling more reliable and efficient mitigation strategies beyond post-processing methods.

To explore these hallucination trends, we analyze models ranging from 70 million to 12 billion parameters within Pythia suite (Biderman et al., 2023), assessing them across various training checkpoints and tasks. Our goal is to validate the oscillatory behavior observed by Li et al. (2024) through evaluation metrics including HaluEval (Li et al., 2023), FactScore (Min et al., 2023), SelfCheck-GPT (Manakul et al., 2023), and XSum (Narayan et al., 2018). Utilizing the reliability of internal model dynamics for quantifying hallucination likelihood, we use EigenScore (Chen et al., 2024) and Semantic Entropy (Kossen et al., 2024) to detect hallucination risk by analyzing variability in high-temperature outputs. Experiments utilize EigenScore and the HELM dataset (Su et al., 2024) to identify hallucinations during training.

In response to this variance, we introduce a novel training protocol called **Sensitivity Dropout** (SenD). SenD is designed to emphasize confident learning of facts, and in turn reduce the likelihood of confabulations, rather than solely minimizing the training loss. By selectively dropping Sensitive Embedding Indices—those exhibiting significant fluctuations throughout training—SenD acts as a technique that reduces hallucination variance and enhances the model's factual certainty. This provides a more reliable criterion for determining training termination, ensuring models not only achieve loss convergence but also display stable factual confidence. To maintain efficiency as model size and inference count increase, we propose the **Efficient EigenScore** (EES), a novel metric for hallucination detection. EES replaces EigenScore (Chen et al., 2024), the primary metric used in our experiments, offering a scalable solution with high correlation to EigenScore.

Our contributions to the field can be summarized as follows, emphasizing that SenD enhances the training process but **does not replace** post-hoc solutions, which may still be required after training:[1]

1. Empirical verification of the **oscillatory nature of hallucinations in LLMs training** across various model scales and detection metrics.

2. **Sensitivity Dropout (SenD)**, a training-time method designed to reduce hallucination variance and increase model factual confidence during training.

3. **Efficient EigenScore (EES)**, an efficient hallucination detection metric used to keep SenD efficient, achieving up to 2x speedup with minimal effects on accuracy.

## 1.2 RELATED WORK

The majority of research on hallucinations in language models has focused on detecting and mitigating this phenomenon rather than explaining its underlying causes. Recent techniques can be categorized into two main approaches: those that rely on output probabilities at inference time (Manakul et al., 2023; Joshi et al., 2017; Li et al., 2023) and those that utilize internal representations or hidden layers of the model (Su et al., 2024; Chen et al., 2024; Kossen et al., 2024). While the former has demonstrated effectiveness, the latter offers deeper insights but often comes with computational trade-offs. Additionally, methods like Reinforcement Learning with Human Feedback (RLHF) have gained traction for enhancing model reliability (Yu et al., 2024). However, many of these post-hoc solutions enhance factual accuracy by layering algorithms atop pre-trained models, which can be inefficient. Our work addresses this gap by focusing on the internal dynamics of the model that contribute to hallucinations.

Regularization techniques have been introduced to fix the issue of variability, notably random neuron dropout, used to reduce the variance and ensure that no neuron is overpowering others (Srivastava et al., 2014; Baldi & Sadowski, 2013). Work such as that done by Santra et al. (2020); Ba & Frey (2013) aims to modify random neuron dropout to change the way neurons are dropped to a more deterministic, precise manner. This has allowed the authors to drop unimportant connections in a deep neural network to ensure that class discriminative information is propagated through the model correctly (Santra et al., 2020). Inspired by this, our aim is to target hallucinatory embedding indices in our models to ensure that factual information is propagated through. State-of-the-art hallucination metrics, especially those based on internal model dynamics, rely on spectral analysis and embedding

---

[1]For the code and datasets used, refer to our GitHub repository at: `https://anonymous.4open.science/r/SeND-Pythia/README.md`.

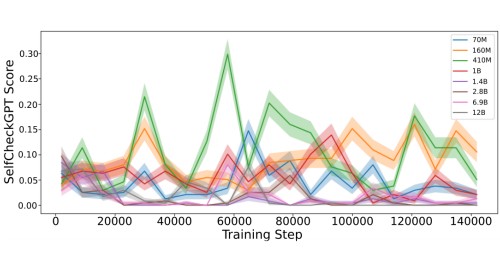

(a) Self-Consistency

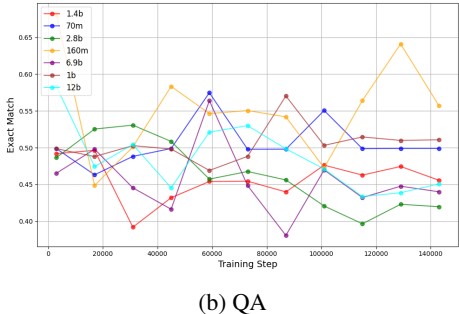

(b) QA

Figure 1: **Visualization of Oscillatory Behavior**. Hallucination metrics are evaluated at equidistant checkpoints of the Pythia models, with sizes 70M, 160M, 410M, 1B, 1.4B, 2.8B, 6.9B, 12B. Part (a) presents the performance of the Pythia models under the SelfCheckGPT metric. Average performance is indicated by solid lines, while the shaded regions represent the standard deviation. Part (b) depicts the same experimental setup, but hallucination measured by the Exact Match (EM) metric of HaluEval. For all model sizes, we observe a pronounced trend of high variance behavior in hallucination rates. This fluctuation emphasizes the need for a mitigation strategy to stabilize performance during training. For Perplexity (PPL), Rouge1 and other HaluEval metrics refer to Appendix A.2.

matrix computations. Methods like EigenScore (Chen et al., 2024) and Semantic Entropy (Kossen et al., 2024) effectively assess hallucination risk but require multiple inferences, making them computationally demanding as models scale. Tools such as the Density of States (DOS) and the kernel polynomial method (KPM) have been explored to approximate spectral properties efficiently (Huang et al., 2023b; Lin et al., 2014). Building on these advancements, our work integrates efficient spectral analysis methods into hallucination detection, demonstrated through EES and SenD.

## 2 OSCILLATORY BEHAVIOUR VALIDATION

The training checkpoints of a transformer model can be vital in understanding the dynamics of how the model learns. Beyond the model's architecture and the data itself, numerous factors influence the learning process such as: whether the loss function penalizes the learner for factual mistakes it makes or if it primarily tries to force the model to memorize the data. While our paper does not aim to address the broader debate on whether LLMs truly understand language or rely on memorization, our analysis of training dynamics through multiple checkpoints shows that converging the training loss does not necessarily correspond to reducing hallucinations, verifying the results by Li et al. (2024) for LLM oscillatory hallucinaiton behaviour during training. Our study utilizes Eleuther AI's Pythia and LMEval tools (Biderman et al., 2023; Gao et al., 2024a) to examine the development and evolution of LLMs throughout the training process. Pythia comprises a suite of 16 LLMs, all trained on public data in the same sequential order, with sizes ranging from 70 million to 12 billion parameters. We use 20 equally spaced training checkpoints from the start to the finish for our analysis. These models are evaluated at each checkpoint on a variety of hallucination metrics such as SelfCheckGPT for Self-consistency (Manakul et al., 2023), XSum for Summarization (Narayan et al., 2018), Perplexity, and HaluEval (Li et al., 2023) for Question Answering (QA) tasks. High SelfCheckGPT Score means that the model is more likely to contradict itself on the given input, a higher Rouge1 score on XSum means the data is aligning better with the provided reference summary, lower perplexity implies higher prediction confidence, and higher Exact Match, Accuracy, and Correctness implies better performance of the model on QA tasks.

### 2.1 HOW DO THE ESTABLISHED ITERATIVE TRAINING PROCESSES INFLUENCE LLM HALLUCINATIONS?

The analysis of hallucination oscillations, as shown in Figure 1, indicates a consistent pattern across different models: oscillations persist throughout training from the initial to the final checkpoint. This finding highlights the uncertainty of halting training solely based on the convergence of training loss. For instance, in QA settings, the optimal Exact Match of the outputs with ground truths is achieved in

earlier checkpoints. This evidence challenges the notion that optimizing solely for unsupervised loss in SGD guarantees learning the most accurate representation of the data. This observation is seen more drastically in 1a, where model size has nearly no effect on the performance of SelfCheckGPT. Instead, we observe oscillatory behaviour within self-consistency, implying that model size is not much effective at tackling the issue of confabulations verified by results in Appendix A.2 as well. Our mitigation approach (SenD) is discussed in Section 4.

## 2.2 How does model complexity affect the emergence of hallucinations throughout training?

An analysis of hallucination detection metrics reveals a diminishing rate of improvement with increased model scaling, particularly up to the 12B parameter size (Appendix A.2 for the study). This suggests that beyond a certain point, even though there is improvement in the hallucinations, larger models do not significantly reduce hallucinations, indicating that scaling alone is not sufficient for building robust models. Instead, more refined approaches are needed to address the underlying variability in model behavior. For the following experiments, we focus on the Pythia 1B model.

## 3 Internal Training Dynamics

Following our investigation of the oscillatory behaviour in training, we look into the internal states of the Pythia 1B model to see what information we are able to extract. In doing so, we define a series of terms and formulas in order to understand the internal processes during the training of LLMs. This information is later used in sections 3.3 and 4 to assist us in deriving methods for improving the variance in the hallucinatory behaviour of models during training.

### 3.1 Sensitive Embedding Indices

To start our analysis of the internal states, we convert the activation matrix of the model into a sentence embedding vector 3.1 which turns an $\mathbb{R}^{n,m}$ activation matrix into a sentence embedding vector $a_k$ for input $k$ with dimension $\mathbb{R}^n$. Given its demonstrated success in hallucination detection by Su et al. (2024), we employ this sentence embedding extraction approach.

**Definition 3.1** (Sentence Embedding Vector). The Sentence Embedding Vector is a way to convert the large $\mathbb{R}^{n,m}$ activation matrix into a smaller, easier to manage vector with dimension $\mathbb{R}^n$.

$$e_k = \frac{1}{2}((\frac{1}{m}\sum_{i=1}^{m} H_{N-1}^i) + H_{N-1}^m) \tag{1}$$

Where $e_k$ is the activation of one input $k$, $m$ is the number of tokens in the sequence, $H$ is the token embedding activation matrix, and $N-1$ is the subtraction to get the penultimate layer index and the formula is adapted from Su et al. (2024). The penultimate layer of the LLM, being the layer closest to the output probabilities, is our primary focus for hallucination analysis due to its rich information about output certainty.

Next, we define the Net Change Formula 3.2 as a way to extract information from the model indicative of oscillatory behaviour between checkpoints from the sentence embedding vector.

**Definition 3.2** (Net Change Formula). Let $e_i^t$ denote the embedding of data point $x$ at embedding index $i$ of the contextual embedding after checkpoint $t$. Then we define the net change formula as

$$\Delta e_i^t = |e_i^t - e_i^{t-1}| \tag{2}$$

With these definitions, we can now describe the crux of our investigation: **Sensitive Embedding Indices (SEIs)**. These SEIs give us key parts of the model that we will prove contribute to the hallucination of LLM models. They can be used to adapt training procedures for lowering hallucination variation during training and better overall confidence at inference time. In essence, SEIs are embedding indices in the sentence embedding from definition 3.1 that experience drastic changes between checkpoints of the training, something we believe is related to the oscillatory behaviour in hallucination performance. When finding the most sensitive embedding indices, we typically want to select the top $K\%$ embedding indices for a specific data point's representation. In our investigation we set $K = 20$.

**Definition 3.3** (Sensitive Embedding Indices - SEIs). Indices of the contextual embedding for data point $x$ which exhibit the highest net change across the last $C$ checkpoints of training, indicating overall high variability during this period. This is calculated by

$$V_i = Var(e_i) \sum_{t=T-C+1}^{T} \Delta e_i^t \quad (3)$$

where $V_i$ is the total variability during the last $C$ checkpoints and the most sensitive embedding indices are

$$\mathbf{s} = \arg \max_{1 \leq i \leq N} \{V_i \mid V_i \geq \text{percentile}(V, 100 - k)\} \quad (4)$$

where N is the embedding vector size and $k$ is the desired percentile threshold.

The above definition of SEIs is then applied to LLM hallucinations through analyses of the Eigen-Scores. In their paper, Chen et al. (2024) define a new metric for detecting confabulations, a subclass of hallucinations. They do this by calculating an EigenScore 3.4 based on determinant calculations from multiple outputs of an LLM with a high-temperature setting (*temperature* set to 0.5) to encourage the LLM to produce a variety of different outputs. They propose that if an LLM is set to hallucinate on that output, the generated texts will show higher semantic variability and produce a higher EigenScore. This method achieves SOTA performance and is unsupervised as it only relies on the representations learned by the model. In the forthcoming sections, we will analyze the correlation between the EigenScore of data points during training checkpoints and the most sensitive embedding indices associated with them.

**Definition 3.4** (EigenScore). The **EigenScore** of data point $x$ indicates the degree of hallucination on input $x$ by the average logarithm of the eigenvalues on the covariance matrix of the multiple output generations (typically 10 in our experiments).

$$ES = \mathbb{E}(Y \mid x, \theta) = \frac{1}{K} \sum_{i=1}^{K} \log(\lambda_i) \quad (5)$$

where $\lambda = \{\lambda_1, \ldots, \lambda_K\}$ denotes the eigenvalues of the regularized covariance matrix $\Sigma + \alpha \cdot \mathbb{I}$. we advise referring to Chen et al. (2024) for a more detailed analysis of this formula.

### 3.2 SENSITIVE EMBEDDING INDEX IMPACT ON EIGENSCORES

To assess the correlation between SEIs and other indices in the embedding matrix of 10 generated outputs at a specific checkpoint, we conduct experiments aimed to determine if the presence of SEIs indicates higher uncertainty and a greater likelihood of hallucinations.

We evaluate the SEI effect on the HELM dataset (Su et al., 2024), which includes outputs from six open-source LLMs based on inference over 50,000 Wikipedia articles, with human annotators labeling passages as factual or hallucinatory. This dataset was selected as Wikipedia is one of the main fact sources people refer to, therefore LLMs should be robust to this type of information as well as it being created with the intended use case of hallucination detection. To assess the impact of SEIs on hallucination, we adapt the EigenScore method by applying it to sentence embeddings from the penultimate layer of EleutherAI's Pythia 1B model, focusing on checkpoints between 133,000 and 143,000 training steps, where embeddings are more stable and the model has a higher degree of language understanding compared to initial checkpoints. We perform SEI dropout, dropping the top 10% of SEIs at each checkpoint, and compare the results to a baseline where 10% of embedding indices are randomly dropped. Additionally, we analyze the impact on hallucination-prone inputs versus non-hallucination-prone inputs to determine if SEIs play a critical role during hallucination, without negatively affecting correct outputs.

#### 3.2.1 WHAT IS THE EFFECT OF SENSITIVE EMBEDDING INDICES ON HALLUCINATION METRICS?

Since a reduction in the EigenScore metric can be used as a proxy to show the reduction in likelihood of hallucination, we keep using this metric in our investigations. We are able to show through our comparison of the baseline random embedding index dropout and SEI dropout that SEIs significantly

reduce the EigenScore metric and in turn, reduce the possibility of a confabulation (Figure 2a), implying their highly important role in determining model's certainty. Not only do we observe this in hallucinatory outputs, we also observe a smaller reduction in EigenScore when applying this technique to correctly answered queries (Figure 2b). This result indicates that our methodology has a significant effect on the uncertainty shown by an LLM. We observe that the internal states of the model are effective in the elimination of confabulating text generation in various model sizes.

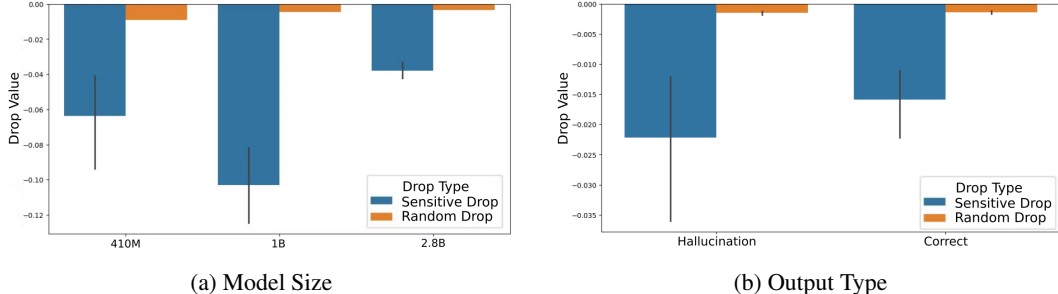

(a) Model Size          (b) Output Type

Figure 2: **Comparison of sensitive embedding index dropout** on inference of Eleuther AI's Pythia various model sizes with random embedding index dropout. (a) Average SEI dropout with standard deviation plotted as scale of the model increases. (b) Average SEI dropout for hallucinatory inputs and non-hallucinatory inputs. Input size for each test is 80 I.I.D. texts. SEI dropping presents a clear, significant reduction in EigenScore compared to that of random embedding index dropping across model sizes. Hallucinatory generations experience a larger drop in EigenScore, meaning that our protocol scales with likelihood of hallucination.

### 3.3 EFFICIENT EIGENSCORE APPROXIMATION

To address the computational complexity of EigenScore calculations, particularly as LLM hidden layer sizes increase, we develop an approximation method. This approximation, detailed in Algorithm 1, leverages the properties of Spectral Density or Density of States (DOS) to estimate EigenScore without explicitly constructing the covariance matrix. While this approximation provides a general overview of EigenScore trends, it is important to note that the output scales differ: EigenScore ranges from $[0, \infty)$, whereas the approximation, referred to as **Efficient EigenScore (EES)**, outputs values between $[-1, 1]$. Since the spectrum of the matrix is altered to make EES computable and operates on its own scale, EES can be seen as a standalone metric for hallucination detection.

The computation of the Efficient EigenScore (EES) is based on two fundamental concepts: Chebyshev Polynomials and Density of States (DOS). A detailed introduction to these concepts is provided in Appendix sections B.1 and B.2. Below, we outline a brief sketch of the derivation of EES. Since Chen et al. (2024) use the covariance matrix of the embedding matrix of 10 generated sequences by the model in their methods, we represent it with $H$ and use it in our derivation.

**Lemma 1.** *Let $f = \log$. Then, for a covariance matrix $H$ with eigenvalues $\lambda_i$, we have*

$$trace(\log(H)) = \sum_{i=1}^{N} \log(\lambda_i), \tag{6}$$

*where $\lambda_i$ are the eigenvalues of $H$.*

**Proposition 1.** *Using the property of the density of states (DOS), we have:*

$$\int \log(\lambda)\,\mu(\lambda)\,d\lambda = \log\left(\prod_{i=1}^{N} \lambda_i\right), \tag{7}$$

*which follows from Lemma 1 since $\sum_{i=1}^{N} \log(\lambda_i) = \log\left(\prod_{i=1}^{N} \lambda_i\right)$.*

Note that from Proposition 1, the integral is equal to $N.EigenScore(H)$ or in our application, given $C$ the integral equals $K.EigenScore(C)$, $K$ being the number of model generations.

---

**Algorithm 1** Efficient EigenScore (EES) Computation Algorithm

---

**Require:** Embedding matrix $E \in \mathbb{R}^{d_{\text{model}} \times K}$, number of Chebyshev terms $M$, number of stochastic trace estimation samples $N_z$

**Ensure:** Approximated EigenScore *EES*

1: **Standardize and Scale the Embedding Matrix $E$:**

2: $E_{\text{mean}} = \frac{1}{K} \sum_{i=1}^{K} E[:,i]$             ▷ Compute mean of $E$

3: $E_{\text{std}} = \sqrt{\frac{1}{K} \sum_{i=1}^{K} (E[:,i] - E_{\text{mean}})^2}$      ▷ Compute standard deviation of $E$

4: $E_{\text{normalized}} = \frac{E - E_{\text{mean}}}{E_{\text{std}}}$             ▷ Standardize $E$

5: $\sigma_{\text{max}} = \text{Power Method}(E_{\text{normalized}})$     ▷ Compute the largest singular value using the power method

6: $E_{\text{normalized}} \leftarrow \frac{E_{\text{normalized}}}{\sigma_{\text{max}}}$           ▷ Scale $E$ by $\sigma_{\text{max}}$

7: **Initialize:**

8: $d_m = 0 \quad \forall m \in \{0, 1, \ldots, M\}$        ▷ Initialize $d_m$ coefficients

9: $c_m = 0 \quad \forall m \in \{0, 1, \ldots, M\}$        ▷ Initialize $c_m$ coefficients

10: **Compute DOS coefficients $d_m$:**

11: **for** $m = 0$ to $M$ **do**

12:      **Sample** $z_j \sim \mathcal{N}(0, I)$      ▷ Sample random vectors for stochastic trace estimation

13:      **Compute Chebyshev polynomial using the recurrence relation**

14: **end for**

15: **Compute Chebyshev coefficients $c_m$:**

16: **for** $m = 0$ to $M$ **do**

17:      $c_m \leftarrow \int_0^1 \log(\lambda) T_m^*(\lambda) \, d\lambda$      ▷ Using Equation 27 and Gaussian Quadrature for approximation

18: **end for**

19: **Compute EigenScore:**

20: $EES \leftarrow \frac{1}{K} \sum_{m=0}^{M} d_m c_m$      ▷ Approximate EigenScore using DOS coefficients

21: **return** *EES*            ▷ Return the approximated EigenScore

---

Our objective is to simplify the integral and approximate its value, avoiding the direct computation of the covariance matrix. This approach is intended to mitigate the computational complexity and associated costs of explicitly handling the covariance matrix. Further utilizing Chebyshev Polynomials, DOS, and KPM (as introduced in Appendix B.2), we can simplify the integral mentioned in Equation 7 to $\sum_{m=0}^{M} d_m c_m$, where $d_m$ term in DOS is approximated using Stochastic Trace Estimation and $c_m$ m'th Chebyshev Polynomial coefficient. Appendices B.3 and B.4 provide the derivation of this equation. Note that the simplified integral is ultimately used to approximate the EigenScore of the matrix which is ultimately equivalent to $\frac{1}{K} \sum_{m=0}^{M} d_m c_m$. Performance of EES approximation is closely correlated with that of the original EigenScore which can be seen in Figure 10.

## 3.4 How does EES scale compared to regular EigenScore?

The efficiency of EES is compared to that of the regular EigenScore calculation with respect to scaling matrix sizes. These tests are imperative to the application of our training protocol on increasing LLM sizes in Section 4 due to larger matrix sizes to decompose for the EigenScore calculation. We conduct a grid search over two important parameters: Matrix size (Figure 3) and Moments used for EES calculation (Figure 9). The difference between EES time in comparison to EigenScore when increasing the number of columns and rows is visualized in Figure 3 using a moments value of 20. It is evident that EES provides a significant computational advantage when increasing the number of columns or rows. Remarkably, at matrix size $\mathbb{R}^{1e8}$, EES nearly halves the computation time of regular EigenScore calculation at around 4 seconds whereas EigenScore takes approximately 7 seconds to calculate. We can then deduce that given a good enough approximation, EES provides a significant reduction in computational complexity as model and matrix size increase.

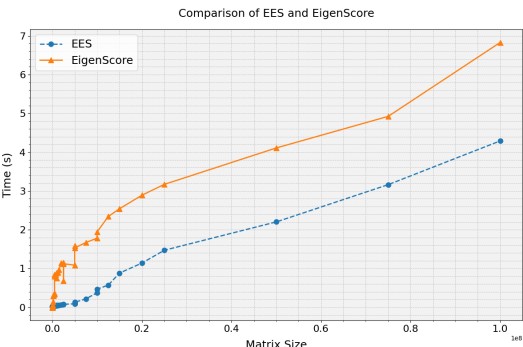

Figure 3: **Efficient EigenScore approximation scaling investigation**. The figure shows the difference in computation time between regular EigenScore calculation and EES with a moments value of 20. The x-axis represents the product of the matrix's rows and columns, and the y-axis shows the computation time. As matrix size increases, EES consistently reduces computation time, making it a practical choice for large LLMs.

## 4 SENSITIVITY DROPOUT (SEND)

Building on the findings from Section 3.2, and aiming to reduce variance in the factual uncertainty of LLMs during training, this section introduces SenD, an efficient and transferable framework for training LLMs. SenD integrates the EES method discussed in Section 3.3 to enhance computational efficiency while addressing variance in SEI behavior. By identifying SEIs, which contribute to the oscillatory behavior of hallucinations during training, SenD deterministically drops these embedding indices based on a small subset of the training data. This approach ensures an increase in the model's factual certainty by the end of training as explained in Algorithm 2.

---

**Algorithm 2** Sensitivity Dropout

---

**Require:** $\epsilon$ denotes the acceptable range for loss convergence and $\delta$ denotes acceptable range for confabulation (EES) convergence
1: Initialize dataset with $\alpha\%$ training $Y_t$ and $(100 - \alpha)\%$ tracking $Y_s$
2: **while** Loss $> \epsilon$ and EES $> \delta$ **do**                          ▷ Refer to Algorithm 1 for EES
3:     **for** $t$ in T **do**                   ▷ T denotes the number of checkpoints per SEI calculation
4:         Train LLM for one checkpoint over $Y_t$
5:         Record penultimate layer representations $R_t$ of LLM over $Y_s$
6:     **end for**
7:     **for** $t \in T - 1$ **do**
8:         Calculate variability $V_t$ between $R_t$ to $R_{t+1}$                          ▷ Refer to Equation 3
9:     **end for**
10:     Take average Variability $V_{avg} = \frac{1}{N_s} \sum_{i=0}^{N_s} V_i$
11:     $s = K$ most sensitive embedding indices $\in V_{avg}$                          ▷ Refer to Equation 4
12:     Drop embedding indices $s$ for next T checkpoints
13: **end while**

---

### 4.1 SEND EXPERIMENT SETUP

To evaluate SenD, we use Pythia 1B model (Biderman et al., 2023), Llama 3.2 1B, and Llama 3.1 8B (Dubey et al., 2024) continuing their training on specific datasets rather than restarting pretraining for efficiency. We continually train the models on the following datasets: HELM, consisting of Wikipedia text (Su et al., 2024), MedHALT, a medical dataset emulating real-world entrance exam questions (Pal et al., 2023), and LegalBench consisting of data for reasoning in LLMs (Guha et al., 2023). Note that HELM and MedHALT are specifically designed for hallucination detection/mitigation in LLMs. SenD is trained on all datasets using 200 data points and and 2,000 points (referred to as 2k) for MedHALT due to the medical domain importance. SenD implements the

EigenScore reduction technique from Section 3.2 and detects SEIs using a 3-checkpoint window on a specialized hallucination tracking dataset. The distance between checkpoints and the dropout rate $K$ are tunable hyperparameters. Given our ablation study in Appendix C, we opt for $K = 20\%$ and Threshold=3 for the experiments. SEIs in the penultimate layer are identified based on their variability across checkpoints and are deterministically dropped for the subsequent 3 training checkpoints. This is repeated at each 3-checkpoint interval until loss convergence, effectively mitigating hallucination tendencies and oscillations.

## 4.2 PERFORMANCE OF SEND ON PYTHIA AND LLAMA MODELS

Pythia and Llama training results are illustrated in Figure 4. To validate that EES accurately approximates the EigenScore metric; we compare the model's progress during training detailed in Appendix B.6. Upon confirming that, we proceed to compare the performance of Pythia 1B, Llama 3.2 1B, and Llama 3.1 8B trained using training without dropout to that of SenD (Figure 4). As shown in the figure and detailed in Appendix D, across all three models and most domains, training with SenD results in a greater reduction in EES during training. In most cases, the final model trained with SenD achieves a lower EES compared to standard training which increases the EES score, demonstrating its effectiveness in reducing hallucination variance and improving the model's final certainty. In addition, there seems to be better improvements using SenD in LegalBench settings. A possible explanation for SenD's superior performance on LegalBench compared to the other two domains which are more general is that early in training, the model sees less domain-specific data, leading to better generalization. However, confirming this hypothesis requires further investigation beyond the scope of this work.

To assess the effectiveness of SenD in comparison to other state-of-the-art factuality metrics, and not solely rely on EES for hallucination detection, , we employ the FactScore metric (Min et al., 2023), which quantifies the factual accuracy of content generated by large language models (LLMs) and HaluEval in Summarization setting (Li et al., 2023). The fact-checking for FactScore is conducted using the HELM dataset on the final trained models (Pythia 1B SenD and 1B Normal Training). A higher FactScore indicates improved factual precision and for HaluEval, higher Exact Match, Accuracy, and Correctness imply better matching with ground truth facts. FactScore is done on 100 and 1000 data points subsequently and table 1 presents the results which imply the better performance of the end model trained with SenD up to 40%. This also depicts the correlation between low EES and high FactScore or HaluEval metrics which are other SOTA metrics for hallucination detection.

Table 1: **Final Model Hallucination Performance: SenD vs. Normal Training (Pythia 1B)**

| Task | Metric | SenD | Normal |
|------|--------|------|--------|
| HaluEval Summarization (LMEval) | Accuracy | **0.016** | 0.014 |
| | Correctness | 0.027 | 0.027 |
| | Exact Match | **0.589** | 0.496 |
| FactScore (100 points) | Score | **0.07** | 0.05 |
| FactScore (1000 points) | Score | **0.08** | 0.06 |

Since SenD is the first method to focus on Hallucinations during the training of LLMs, there are no baselines or SOTA methods to compare it to. However, one could treat SenD as a post-hoc method and compare it to RAG (Lewis et al., 2021) to get a sense of how different training based methods compare to in-context methods. When tested on 1000 data points, SenD-continually-trained Pythia achieves a FactScore of 0.07 as in Table 1, while the base Pythia with RAG scores 0.25, highlighting RAG's effectiveness in reducing hallucinations by providing context. However, to make the comparison fair, applying RAG to a SenD-trained model achieves a higher FactScore of 0.28, outperforming RAG on a normally trained model by 12%. This indicates that even though SenD is not meant to outperform post-hoc methods, SeND combined with RAG enhances the end model's factual performance compared to RAG on a normally trained model.

## 5 CONCLUSION & FUTURE WORK

In this paper, we presented a protocol to refine the current training methods of LLMs based on experiments showing oscillatory behaviour with respect to hallucinations throughout training (Figure 1).

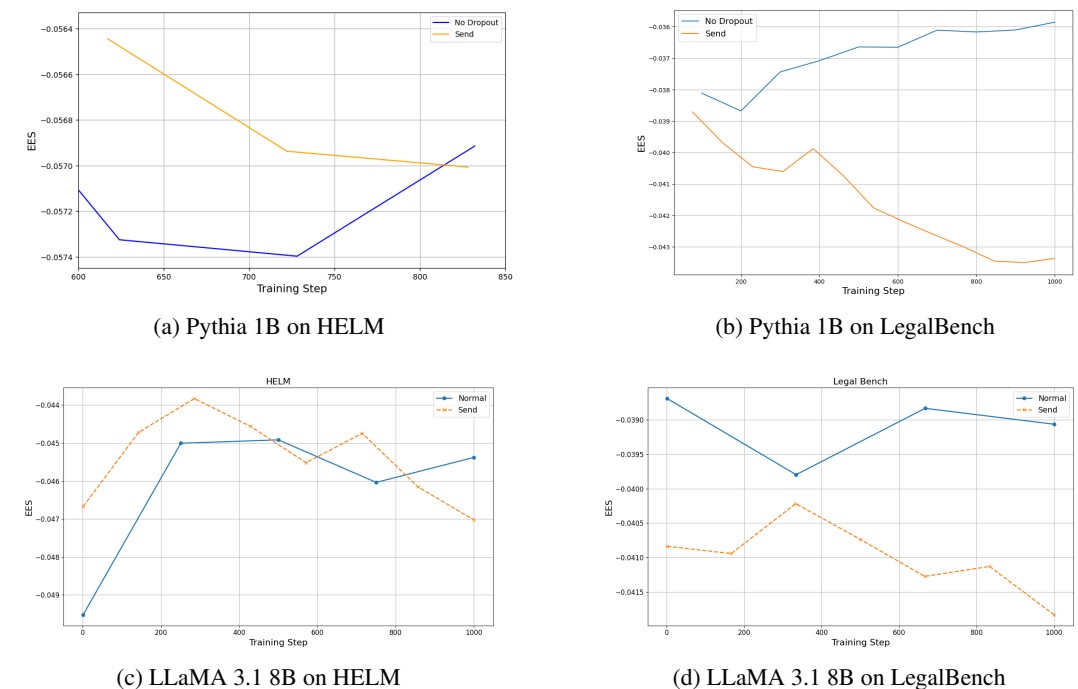

(a) Pythia 1B on HELM          (b) Pythia 1B on LegalBench

(c) LLaMA 3.1 8B on HELM       (d) LLaMA 3.1 8B on LegalBench

Figure 4: **Regular Training vs. SeND on HELM and LegalBench datasets.** The first row shows results of training Pythia 1B on (a) HELM and (b) LegalBench and the second row for Llama 3.1 8B. In all cases, training with SeND demonstrates a more controlled reduction in **EES**, optimizing for hallucination mitigation and loss stability. Results are averaged over 5 runs, and models are trained until loss convergence with $K = 20\%$ and Threshold = 3. For results on the Helm dataset and Llama 3.2 1B training, refer to Appendix D

To do this we used the internal states of LLMs, specifically the penultimate layer activations during inference on a specialized dataset. We present an initial method of reducing hallucinations based on the principles of EigenScore metrics introduced by Chen et al. (2024). We showed empirically that our SEI detection method significantly reduces the EigenScore on inference of LLMs throughout various stages of training (Figure 2). Following the success of the SEI method, we moved on to the application of a hallucination reduction method on training on Pythia and Llama models and various domains. We show through training with SenD that we are able to fix the oscillatory behaviour initially seen throughout training and reduce the EES of finetuned models as shown in Figure 4 by modifying the internal mechanics of training with **Sensitivity Dropout**. At test time we achieve a 40% increase in FactScore performance and improvement of other SOTA hallucination detection metrics, verifying that SenD provides a substantial improvement to current training protocols both during and after training.

Note that although SenD has only been applied to finetuning training in this paper, the training framework is applicable to all stages of training. We encourage future work to implement SenD on larger training sets, such as pretraining, to see how SenD performs in these environments. To further advance our work, we plan to scale SenD to larger datasets and models, as current experiments were limited by compute constraints with larger LLMs. Demonstrating SenD's effectiveness on larger open-source models like Meta's LLaMA 3.2 405B (Dubey et al., 2024) will provide crucial evidence for organizations developing state-of-the-art LLMs to incorporate SenD into their training protocols, ultimately improving model safety. Given that SenD targets variance reduction during training, we anticipate even greater gains on larger LLMs, where the higher inherent variance may amplify the regularization effect and lead to more significant improvements.

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

## A    ADDITIONAL EXPERIMENTS

### A.1    DRASTIC EMBEDDING CHANGES LEADING TO SENSITIVE EMBEDDING INDICES

Looking at internal states of the model allows getting a deeper understanding of the dynamics that could be leading to the oscillatory behaviour seen in Figure 1. To do this, we record the net change (Definition 3.2) between checkpoints of the penultimate layer where one checkpoint would be the correct answer and the next would hallucinate. This net change with respect to various different input texts is plotted in Figure 5. It can be observed that there were specific embedding activations that experienced drastically more change relative to the rest of the embeddings. This is the main source of motivation to further define SEIs (Definition 3.3).

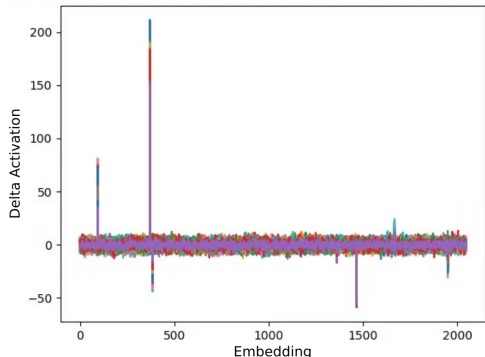

Figure 5: **Net change of sentence embeddings** between checkpoints 125,000 and 143,000. Each different colour is a different input text. As depicted, there are specific embedding indices that go through drastic changes between the two checkpoints of the training regardless of the input.

## A.2 HALLUCINATION OSCILLATIONS ACROSS MODEL SIZES

Figures 6, 7, and 8 show our study of hallucination oscillations during the training of Pythia models. An overall observation across the plots is that as opposed to our intuitive expectation which is a linear decrease of the hallucination detection metric when the model scales linearly, neither the oscillations during the training of the model decrease, nor the end model reaches its optimal state in terms of the hallucination metric.

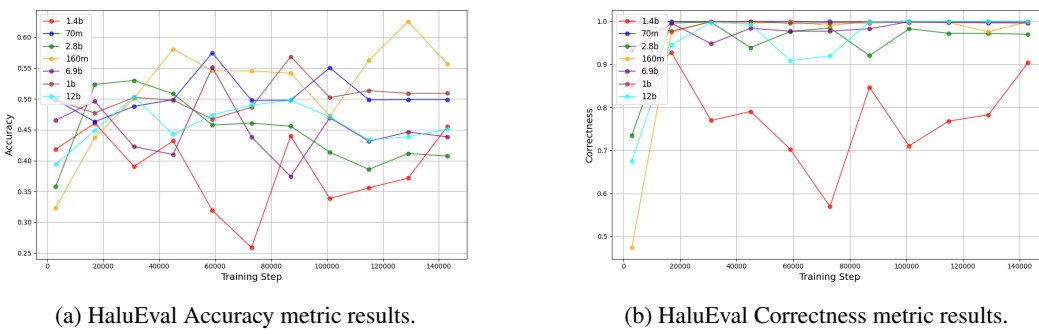

(a) HaluEval Accuracy metric results.     (b) HaluEval Correctness metric results.

Figure 6: Ablation studies on various HaluEval metrics for hallucination detection on Pythia suite.

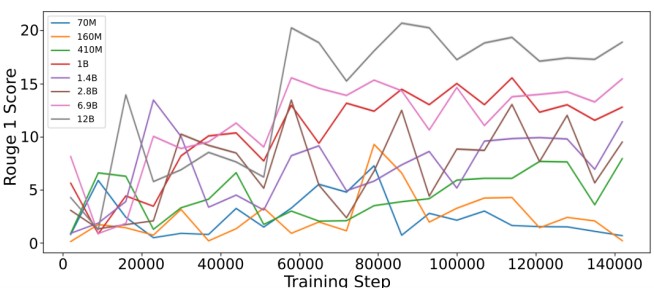

Figure 7: XSum Rouge 1 Score metric results on Pythia suite.

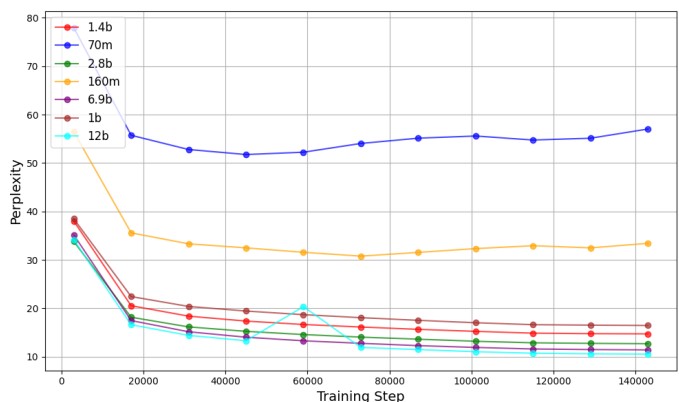

Figure 8: Perplexity (PPL) metric results on Pythia suite.

# B  EFFICIENT EIGENSCORE (EES) DERIVATION

## B.1  BACKGROUND: CHEBYSHEV POLYNOMIALS

Chebyshev polynomials are a sequence of orthogonal polynomials in the interval $[-1, 1]$ – orthogonality property shown in equation 8 – that are widely used in numerical analysis, approximation theory, and other areas of applied mathematics. In this work, we are mainly concerned with the Chebyshev polynomials of the first kind with the recurrence relation shown in equation 9. Note that this recurrence could also be applied to matrices. Any function $f$ defined in the interval $[-1, 1]$ can be approximated with the Chebyshev expansion as shown in 10.

$$\int_{-1}^{1} \frac{2}{(1 + \delta_{0n})\pi\sqrt{1 - x^2}} T_m(x) T_n(x) \, dx = \delta_{mn},$$

$$\text{where} \quad \delta_{mn} = \begin{cases} 1 & \text{if } m = n, \\ 0 & \text{if } m \neq n, \end{cases} \tag{8}$$

$$\begin{aligned} T_0(x) &= 1, \\ T_1(x) &= x, \\ T_{n+1}(x) &= 2x \cdot T_n(x) - T_{n-1}(x), \quad \text{for } n \geq 1. \end{aligned} \tag{9}$$

$$f(x) = \sum_{n=0}^{\infty} c_n T_n(x), \tag{10}$$

$$\text{where } c_n = \frac{2}{\pi} \int_{-1}^{1} \frac{f(x) T_n(x)}{\sqrt{1 - x^2}} \, dx \text{ for } n > 0, \tag{11}$$

$$c_0 = \frac{1}{\pi} \int_{-1}^{1} \frac{f(x)}{\sqrt{1 - x^2}} \, dx. \tag{12}$$

## B.2  BACKGROUND: DOS AND KPM

Let $H$ be a symmetric matrix $H \in \mathbb{R}^{N \times N}$ with an eigendecomposition $H = Q\Lambda Q^T$, where $\Lambda = \text{diag}(\lambda_1, \cdots, \lambda_N)$ and $Q = [q_1, \cdots, q_N]$ is orthogonal. The spectral density induced by $H$ is the generalized function:

$$\mu(\lambda) = \frac{1}{N} \sum_{i=1}^{N} \delta(\lambda - \lambda_i), \tag{13}$$

where $\delta$ is the Dirac delta function. For any analytic test function $f$, the integral of $f$ with respect to $\mu$ is:

$$\int f(\lambda)\mu(\lambda)\,d\lambda = \text{trace}(f(H)). \tag{14}$$

Dong et al. (2019) introduced KPM as a numerical technique to approximate DOS. KPM approximates DOS by expanding it in terms of chebyshev polynomials. Requiring the matrix's spectrum to be supported in the interval $[-1, 1]$, KPM approximates DOS with the following formula, $\lambda$ being the eigen value of the matrix $H$ and $d_m$ approximated by Stochastic Trace Estimation:

$$\mu^{\approx}(\lambda) = \sum_{m=1}^{\infty} d_m T_m^*(\lambda), \tag{15}$$

$$\text{where} \quad d_m = \frac{1}{N}\text{trace}(T_m(H)), \tag{16}$$

$$\text{and} \quad d_m \approx \frac{1}{N}\frac{1}{N_z}\sum_{j=1}^{N_z} \mathbf{z}_j^T T_m(H)\mathbf{z}_j, \tag{17}$$

$$\text{and} \quad T_m^*(x) = \frac{2}{(1+\delta_{0m})\pi\sqrt{1-x^2}}T_m(x). \tag{18}$$

In the application for hallucination detection, we can use equation 14 to derive a formula for the EigenScore approximation using the properties of Chebyshev polynomials and DOS.

### B.3 STOCHASTIC TRACE ESTIMATION ON EMBEDDING MATRIX

We are interested in computing the $d_m$ term of DOS relying solely on the embedding matrix $E$ therefore we need to rewrite $d_m$ as follows:

$$d_m = \frac{1}{K}\frac{1}{N_z}\sum_{j=0}^{\infty} z_j^T T_m(E^T E)z_j \tag{19}$$

where $T_m$ can be computed using the Chebyshev polynomials of matrix $C = E^T E$.

$$T_0(E^T E)\mathbf{z}_j = I\mathbf{z}_j = \mathbf{z}_j,$$
$$T_1(E^T E)\mathbf{z}_j = E^T E\mathbf{z}_j,$$
$$T_{m+1}(E^T E)\mathbf{z}_j = 2E^T E T_m(E^T E)\mathbf{z}_j - T_{m-1}(E^T E)\mathbf{z}_j$$

Each term can be computed with a matrix-vector multiplication.

### B.4 EES INTEGRAL CALCULATION

Given the orthogonality of the Chebyshev polynomials, we can simplify the integral mentioned in proposition 1. To approximate the EigenScore, we will expand $\log(\lambda)$ in terms of Chebyshev polynomials and use their orthogonality to simplify the integral.

**Expanding and Integrating**

To approximate the integral:

$$\frac{1}{K}\int \log(\lambda)\mu(\lambda)\,d\lambda \tag{20}$$

Substitute the Chebyshev Expansion for DOS:

$$\mu(\lambda) \approx \sum_{m=0}^{M} d_m T_m^*(\lambda) \tag{21}$$

where:

$$T_m^*(\lambda) = w(\lambda)T_m(\lambda) = \frac{2}{\pi\sqrt{1-\lambda^2}(1+\delta_{0m})}T_m(\lambda)$$

Distribute $\log(\lambda)$ in the integral:

$$\frac{1}{K} \int \log(\lambda) \left( \sum_{m=0}^{M} d_m T_m^*(\lambda) \right) d\lambda = \frac{1}{K} \sum_{m=0}^{M} d_m \int \log(\lambda) T_m^*(\lambda) \, d\lambda \tag{22}$$

**Evaluate the Integral Using Orthogonality:**

To simplify the integral,

$$\int \log(\lambda) T_m^*(\lambda) \, d\lambda \tag{23}$$

First, express $\log(\lambda)$ as a series of Chebyshev polynomials:

$$\log(\lambda) = \sum_{m=0}^{\infty} c_m T_m(\lambda) \tag{24}$$

Then:

$$\int_0^1 \log(\lambda) T_m^*(\lambda) \, d\lambda = \int_0^1 \left( \sum_{m=0}^{\infty} c_m T_m(\lambda) \right) T_m(\lambda) \, d\lambda \tag{25}$$

$$\tag{26}$$

Note: The lower bound of the integral is 0 as the matrix is defined in the spectrum $[0,1]$.

Using the orthogonality, we get:

$$c_m = \int_0^1 \log(\lambda) T_m^*(\lambda) \, d\lambda \tag{27}$$

So the integral simplifies to:

$$\frac{1}{K} \sum_{m=0}^{M} d_m c_m \tag{28}$$

### B.5 EFFICIENT EIGENSCORE MOMENTS

Figure 9 presents the effect of using different moment values as the number of matrix rows increases with respect to time. This is an important hyperparameter to tune as increasing the number of moments on EES correlates to having a more accurate and representative approximation of the EigenScore. We observe that as moments in EES increase, the time to calculate EES increases. From this result, we conclude that selecting a moment value of under 50 would provide a balanced trade-off between accuracy and calculation time.

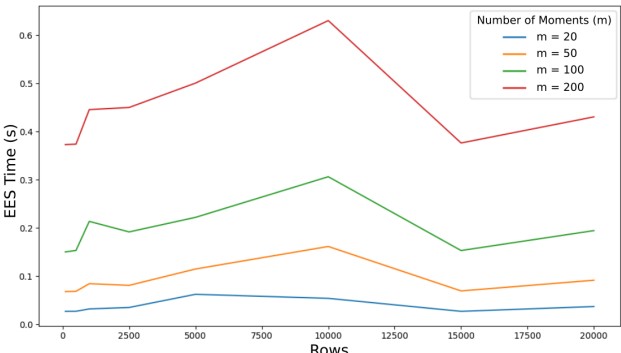

Figure 9:
Effect of changing number of moments on EES calculation time (seconds). More moments gives more accurate approximation but higher computation time.

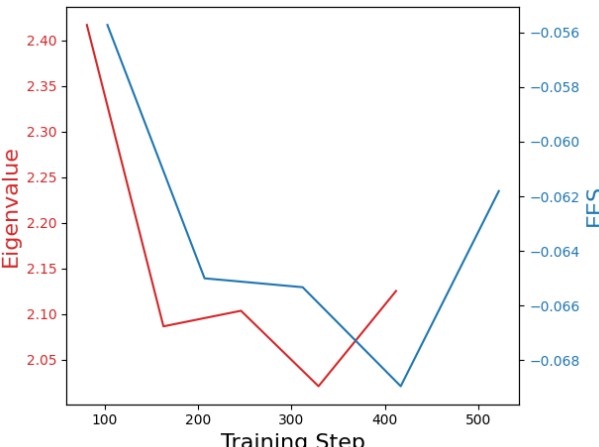

Figure 10: Performance of SenD on Pythia 1B wih HELM dataset computed with both EES and regular EigenScore. EES is able to closely track the true EigenScore performance metric, showing that it is a good approximator.

### B.6 EIGENSCORE AND EES TRAINING TRAJECTORIES

To demonstrate the high correlation between EigenScore and EES, we record the progress of Pythia 1B finetuning on the HELM dataset using both EigenScore and EES hallucination metrics (Figure 10). Albeit a different scale and window, the trajectories, magnitude and shape of the graphs are nearly identical while EES takes only 4 minutes to calculate and EigenScore takes approximately 8, an astounding 2x increase in compute speed. These results show that our metric closely resembles the target metric while greatly reducing the required computational resources.

## C ABLATION STUDY ON $K$ AND STEP THRESHOLDING FOR SEND

Figure 11 shows the ablation study done on $K$ and Figure 12 illustrates the ablations study done on the Step Threshold for SenD experiments. As depicted, $K = 20\%$ and Threshold = 3 are chosen for our experiments except for Llama 3.1 8B model which due to its larger size requires more embedding indices to be dropped, hence adapting to $K = 30\%$.

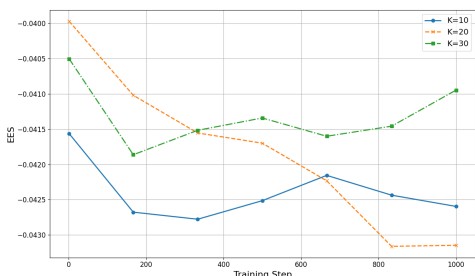
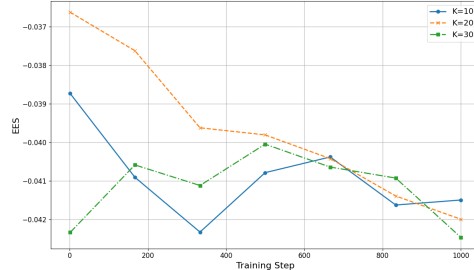

(a) Ablation study on K using the LegalBench dataset.

(b) Ablation study on K using the MedHalt dataset.

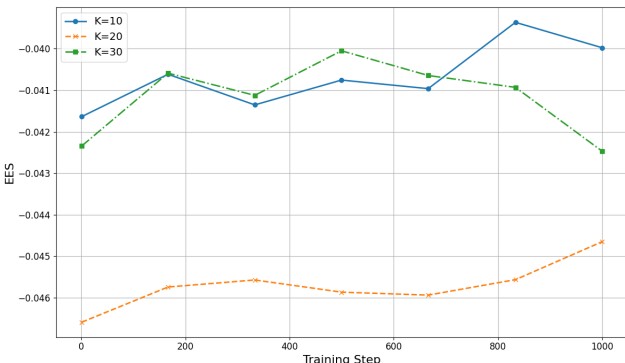

(c) Ablation study on K using the HELM dataset.

Figure 11: Ablation on dropout rate $K \in \{10\%, 20\%, 30\%\}$ using the Pythia 1B model averaged over 10 runs on the LegalBench dataset. $K = 20\%$ achieves optimal performance in reducing EES throughout training for HELM and LegalBench and almost equalizes $K = 30\%$ in stabilizing the halluciantion oscillations, therefore we choose $K = 20\%$ for our experiments.

## D  ADDITIONAL PYTHIA 1B, LLAMA 3.2 1B, AND LLAMA 3.1 8B TRAINING WITH AND WITHOUT SEND

Here, we present additional experimental results of training Pythia and LLaMA on multiple domains. Figure 14b supplements the results discussed in Section 4 by illustrating the training procedures on the Medical domain.

In the Pythia 1B setting, the EES achieved with training using SeND remains consistently lower than that of normal training and exhibits fewer oscillations throughout the training process. In the LLaMA 3.1 8B setting, while both approaches show an increase in the EES metric during training, the final model trained with SeND achieves a lower EES, indicating a reduced likelihood of hallucinations in this domain.

Figure 14 visualizes the training results for the LLaMA 1B model across all domains. Similar trends to the other experimental settings are observed. In the HELM and LegalBench settings, the EES of the model trained with SeND shows a significant reduction by the end of the training. However, in the MedHalt setting, this reduction is less pronounced by the conclusion of training.

Figure 15 depicts the standard deviations of 5 runs experimented on the Pythia 1B model. As shown, the error shading area is not too wide hence proving the reliability of the results.

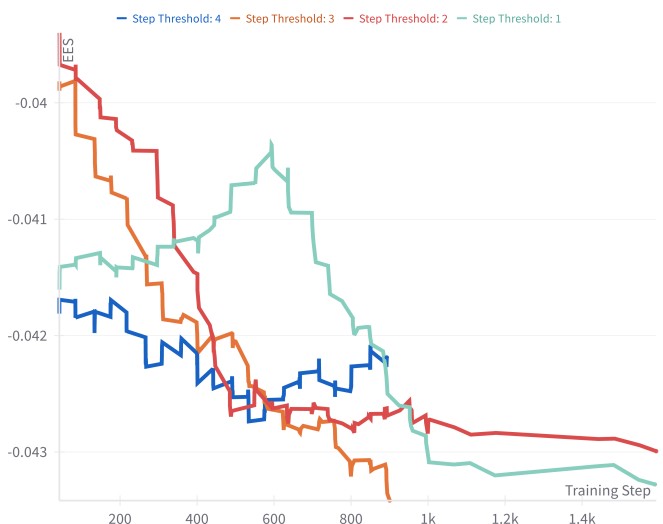

Figure 12: Ablation on Step Threshold $\in \{1, 2, 3, 4\}$ on the Pythia 1B model with the LegalBench dataset. The fastest drop in EES is achieved by setting Threshold = 3, therefore we choose Threshold = 3 for our experiments. Results are averaged over 5 multiple runs.

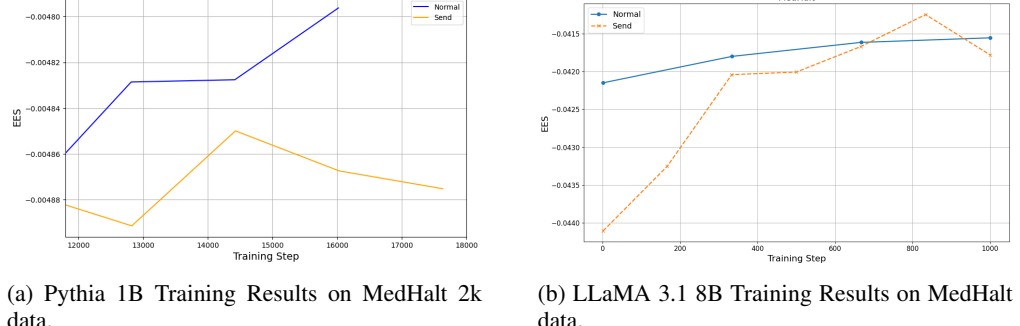

(a) Pythia 1B Training Results on MedHalt 2k data.

(b) LLaMA 3.1 8B Training Results on MedHalt data.

Figure 13: Comparison of Training Results on the MedHalt data: (a) Pythia 1B, (b) LLaMA 3.1 8B both averaged on 5 runs.

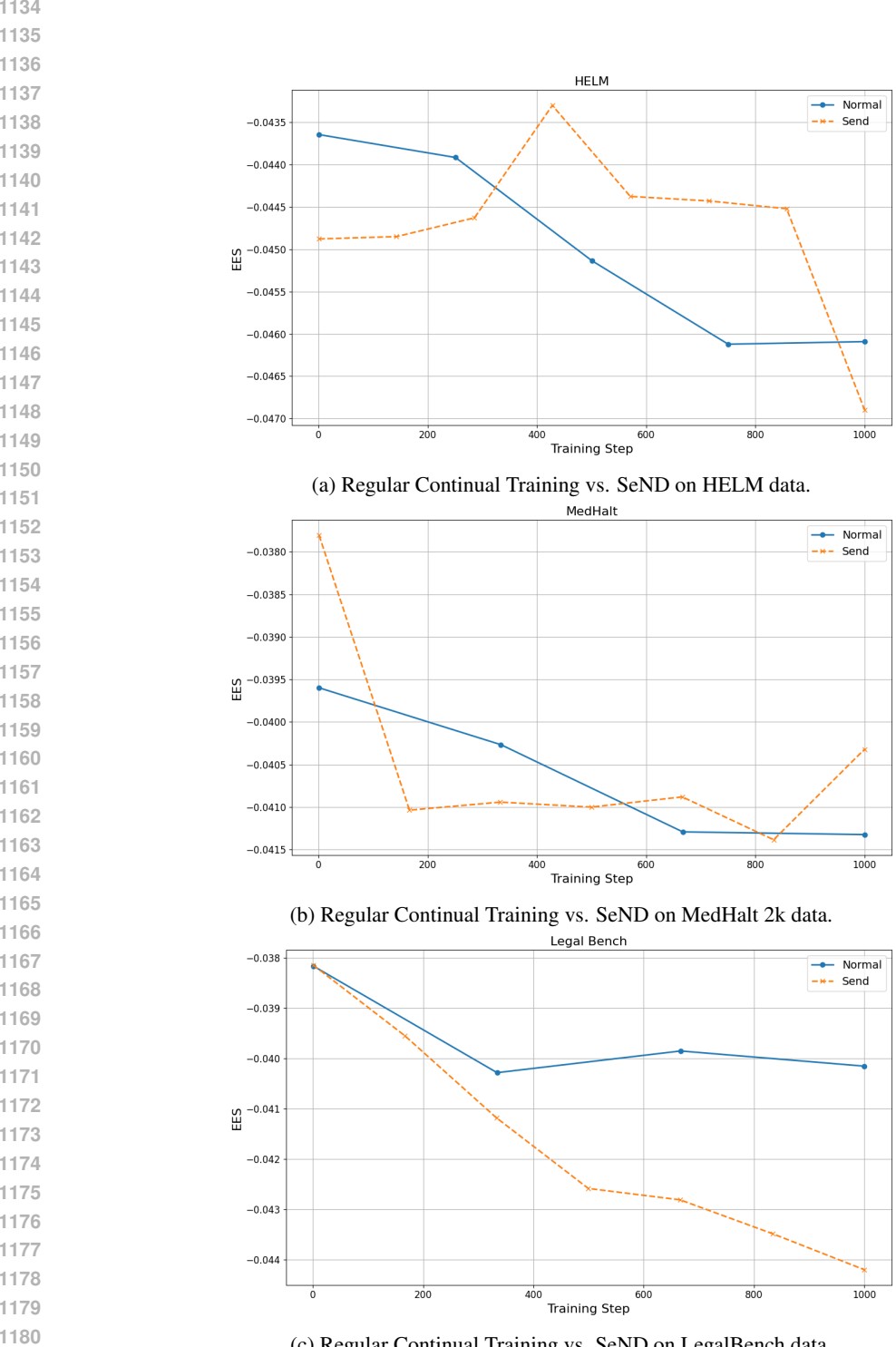

(a) Regular Continual Training vs. SeND on HELM data.

(b) Regular Continual Training vs. SeND on MedHalt 2k data.

(c) Regular Continual Training vs. SeND on LegalBench data.

Figure 14: LLaMA 3.2 1B Training Results: Comparison of Regular Continual Training and SeND on the HELM, MedHalt, and LegalBench datasets averaged over 5 runs. SeND consistently outperforms Regular Continual Training in reducing hallucinations and stabilizing EES across all datasets.

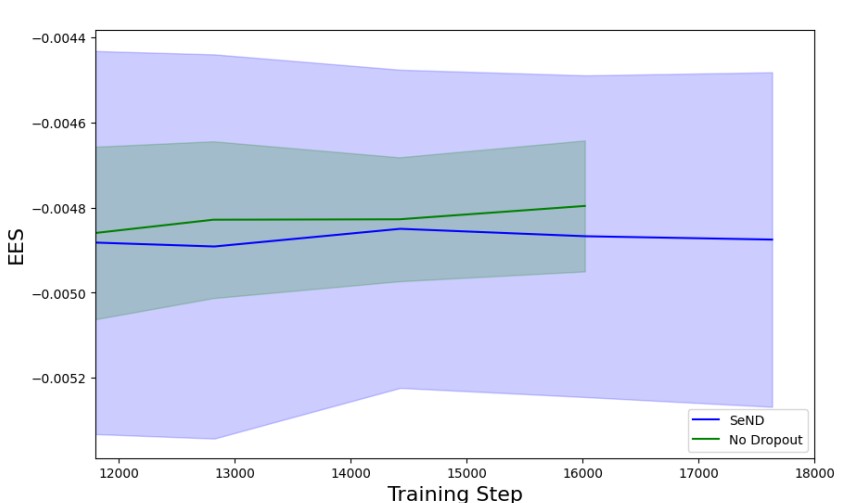

(a) Pythia 1B SenD vs. Normally trained on MedHalt dataset.

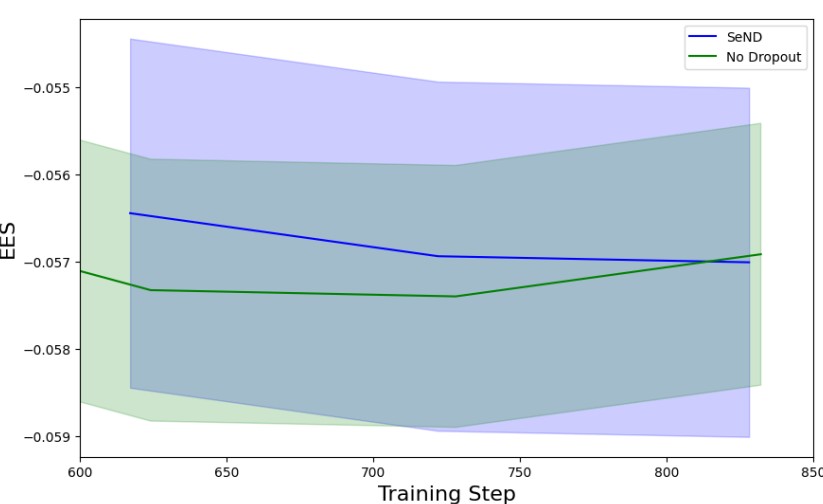

(b) Pythia 1B SenD vs. Normally trained on HELM dataset.

Figure 15: Standard deviation (error) analysis for 5 runs of Pythia 1B training on HELM and Med-Halt domains using SenD and normal training.

