# OpenReview forum: "Hallucination Detox: Sensitive Neuron Dropout (SeND) for Large Language Model Training"
_ICLR.cc/2025/Conference — Submitted to ICLR 2025_

### Official Review · Reviewer_97yW · 2024-10-22

**Soundness:** 1
**Presentation:** 2
**Contribution:** 1
**Rating:** 5
**Confidence:** 4

**Summary:**

This paper empirically validates the oscillatory nature of hallucinations during the training process of LLMs, despite being discovered by previous work. Subsequently, this paper introduces Sensitive Neuron Dropout (SeND), a training-time method for hallucination reduction, and  Efficient EigenScore (EES), a more efficient hallucination detection metric.

**Strengths:**

1. Different from existing post-hoc detection and mitigation strategies, this paper focuses on the relationship between the training process and the emergence of hallucinations, trying to provide interpretation from a relatively new perspective.
2.  Before introducing the specific method, this paper conducts motivational experiments, consolidating the rationale of the method.

**Weaknesses:**

1. Despite focusing on hallucination, this paper does not test on any hallucination dataset and metrics directly, but utilizes some indicative alternatives. This reduces the credibility of the results.
2. Despite the claims of reduced hallucination, it's unclear whether this technique would hinder the performance of models.
3. For LLMs, it's rare to train models for several epochs to prevent overfiting and catastrophic forgetting, while it seems that this method can only be used for multi-epoch training settings.
4. Xsum is not suitable to evaluate hallucination of LLM, and Rouge1 score is a bit out-of-date/ineffective to evaluate the performance of LLMs.
5. The writing for "Sec. 1.2 Related Work" is quite strange. Here, the 2nd and 4th paragraphs focus on motivation and implementation details instead of the comparison with peer methods.
6. In Sec. 2.2, I observed that generally, the metrics change positively with the increase of LLM sizes, inconsistent with the observations of authors. Could you please provide further explanation?
7. The number of models and datasets is too small (i.e., only 1) to validate the robustness of the method.
8. There is a lack of baseline and performance comparison with post-hoc solutions.

**Questions:**

1. For line 227, the meaning of H needs to be further explained.
2. For line 229, a citation is needed to support the claim.
3. Based on my understanding, Sensitive Neurons refers to specific indices, rather than neurons. This name could be misleading.
4. For line 284, the details of removing operation are unclear.

---

> ### Author Response · Authors · 2024-11-22
> **Author Response to Reviewer 97yW**
>
> We thank the reviewer for mentioning the novelty of the work. In response to the limitations discussed, we have conducted additional experiments as follows:
>
> Weaknesses:
>
> 1. **Hallucination Benchmarks:** In terms of the datasets used, HELM and MedHalt are both **hallucination benchmarks** which are from two disjoint domains. All our experiments for SenD are done using these two hallucination based datasets and we have now added a new domain, **LegalBench**, to satisfy the call for more robust testing (**Section 4.1**).
>
>     For the metrics used we refer the reviewer to the **General Response and Table 1**.
>
> 2. Assuming that “performance” refers to the hallucinations of the model, the performance of the model with respect to hallucination is documented in **Figure 4** as well as in **line 461 and Table 1**. In brief, the model trained with SenD achieves better hallucination performance. On the other hand, if by “performance”, the training loss convergence is intended, we attach below the loss curves for **Llama 3.2 1B with and without SenD**. For **all models**, we train until convergence of loss on the training set. Note that the training loss curves are nearly identical, highlighting that SenD is able to address issues surrounding hallucination, it has no significant, if any, impact at all on the training loss of the models. Figures: https://ibb.co/4Pcr45K and https://ibb.co/KrY6jvf .
> 3. We thank the reviewer for raising the potential for confusion with the use of **epochs** as **checkpoints** for training. This has now been addressed in the paper in **Section 4.1** and **Algorithm 2** and the terminology “epoch” has been replaced by “checkpoint” since even in our experiments, we do single-epoch training and the intention for mentioning this initially was to differentiate between a set of back-to-back training steps.
> 4. As suggested, we have included **Perplexity** and **HaluEval** for these evaluations and more details can be found in the General Response. In summary, the HaluEval QA exhibits the same pattern of oscillatory behaviour as found with previous metrics (refer to **Figure 2b**). Results from Perplexity over training (**Figure 8**) highlight the diminishing returns from increasing model size.
> 5. We value the feedback surrounding Section 1.2 Related Work. We agree that **paragraph 2** would be better suited in the motivation section and have implemented this change such that the metrics used for Evaluation are now in **Section 1.1**. **Paragraph 4** has also been modified to reflect this change.
> 6. Thank you for giving us the opportunity to clarify this. An overall observation across the plots is that, as opposed to our intuitive expectation, neither the oscillations during the training of the model nor the reduction of the hallucination metrics reduce significantly but follow a slow improvement trend which is also observable in the Perplexity metric in **Figure 8**. This is not enough to conclude that scaling alone will solve the issue of high variance in hallucination during training. We argue that to ameliorate this situation, continued scaling will cease to provide benefits and new methods, such as SenD, will be required to provide measurable improvements.
> 7. This has been a common comment from the reviewers and we appreciate the feedback to provide a more sound argument. In response to this, we have outlined our changes in the General Response. In brief, we have increased both the number of models to **3** (**Pythia 1B, Llama 3.2 1B, and Llama 3.1 8B**) as well as an increase in the dataset size (**HELM, MedHalt, and now LegalBench**) all of which are tailored to hallucination tasks.
> 8. We appreciate the mention of comparison to baselines as this has been a common comment from reviewers and has also been addressed in the **General Response**. In brief, SenD is compared to RAG in performance but there is a difference in the functionality which is explained in the General Response. We would like to highlight again that RAG augmented with SenD achieved 12% higher performance than RAG.
>
> Questions
>
> 1. H is the activation matrix where each element is a token embedding vector in the given layer. This has been taken into account in the new revision in **line 197**.
> 2. This is cited from the paper: https://arxiv.org/abs/2403.06448. This is now taken into account in the new revision in **line 199**.
> 3. We greatly appreciate the suggestion for a change here. We have now implemented this change to refer to **embedding indices** instead of **neurons**. For more information we refer the reviewer to the General Response.
> 4. In this case, “removing” refers to **dropping** the indices or setting them to zero. This has been changed throughout the paper from **“remove” to “drop”** to reflect the dropout style procedure that inspired this operation.
>
> We hope to have answered the questions and are happy to discuss any remaining concerns.

---

> ### Comment · Reviewer_97yW · 2024-11-26
> **Response to Authors**
>
> I still have the following question. Could you please further clarify them?
>
> Weekness 2.1: Performance means the accuracy on regular benchmarks, such as GSM8K, MATH, MBPP, HumanEval. Even of roughly positive relationship between loss and accuracy, they are different. Loss cannot sensitively reflect the impact on performance, especially for reasoning tasks.
> Weekness 2.2: The convergence loss of SeND seems higher and less stable than that of normal training. Could you please explain more about it, and provide the loss numbers?

---

> > ### Author Response · Authors · 2024-11-27
> > **Author Response to Reviewer 97yW**
> >
> > Thank you for for your time and valuable feedback. We ran additional evaluations and the results are clarified below:
> >
> > 1. Weakness 2.1:
> >
> >     It is a valid point that the loss is not the sole metric to report the performance of a model on specific tasks. Since our focus is not on achieving state-of-the-art performance but on reducing hallucination variance, we initially did not report loss values in detail and used training loss as a proxy for performance, as it suffices to note that the loss remains consistent with normal training (found in the second table in the response to Weakness 2.2).
> >
> >     As you suggested, to demonstrate that the performance of the model trained with SeND does not degrade compared to the normally trained model, we ran additional evaluations on MMLU and GSM8K. Given the limited dataset size (2,000 points), model size (1B parameters), and the lack of fine-tuning for specific tasks, the tasks mentioned remain challenging to evaluate in this setup. We leave further exploration of more expressive SeND-trained models on these benchmarks to future works. The results of our evaluations on Pythia 1B model are as follows:
> >
> >     | Benchmark/Task | Group | Version | Filter | n-shot | Metric | Value (Normal) | Stderr (Normal) | Value (SeND) | Stderr (SeND) |
> >     | --- | --- | --- | --- | --- | --- | --- | --- | --- | --- |
> >     | MMLU | Overall | 2 | none | - | Accuracy (↑) | 0.2330 | ±0.0036 | 0.2309 | ±0.0036 |
> >     |  | Humanities | 2 | none | - | Accuracy (↑) | 0.2417 | ±0.0062 | 0.2427 | ±0.0062 |
> >     |  | Other | 2 | none | - | Accuracy (↑) | 0.2440 | ±0.0077 | 0.2443 | ±0.0077 |
> >     |  | Social Sciences | 2 | none | - | Accuracy (↑) | 0.2229 | ±0.0075 | 0.2158 | ±0.0074 |
> >     |  | STEM | 2 | none | - | Accuracy (↑) | 0.2192 | ±0.0073 | 0.2147 | ±0.0073 |
> >     | GSM8K | - | 3 | flexible-extract | 5 | Exact Match (↑) | 0.0152 | ±0.0034 | 0.0100 | ±0.0034 |
> >
> >     These results confirm that the performance of models trained with and without SeND is comparable.
> >
> >
> > 1. Weakness 2.2:
> >     - Assuming that **“Loss” refers to EES**, while SeND is not integrated into the training loss function or optimization algorithm like traditional regularization techniques, it nonetheless consistently improves the behaviour of the EES curve during training as shown empirically. Specifically, SeND does not guarantee convergence of EES but significantly reduces variance and final EES values in most cases, as illustrated in **Figures 4, 13, and 14**.
> >
> >         For instance, in **LegalBench** settings (**Figure 4**), SeND results in a smoother EES decline when applied to the Pythia 1B model and achieves clearly superior performance compared to standard training on the LLaMA 3.1 8B model. Similarly, on the **HELM** dataset (**Figure 4**), normal training tends to increase the EES by the end of training and exhibits higher variance. In contrast, SeND consistently reduces the EES throughout training while maintaining lower variance.
> >
> >         Notably, in training the **LLaMA 3.2 1B model** (**Figure 14**), the benefits of SeND are more pronounced in HELM and LegalBench settings. However, the benefits of SenD are less evident on the MedHalt dataset.  We hypothesize that the more oscillatory results of SenD on the HELM dataset in **Llama 3.2 1B** (**Figure 14a**) stem from Wikipedia data dominating its training set compared to the MedHalt and LegalBench. We believe applying SeND earlier in training, before much HELM type data is seen, could improve performance. Across the **majority** of scenarios, SeND leads to lower EES metrics and reduced variance by the end of training. The table below summarizes the final EES values for each training configuration, as requested and the **lower EES** values are highlighted in **bold**:
> >
> >         | Benchmark | Model Name | Model Size | SenD Final EES | Normal Final EES | SenD Final Loss | Normal Final Loss |
> >         | --- | --- | --- | --- | --- | --- | --- |
> >         | HELM | Pythia | 1B | ***-0.0570*** | -0.0569 | 0.02 | 0.02 |
> >         |  | Llama 3.2  | 1B | ***-0.047*** | -0.046 | 0.01 | 0.01 |
> >         |  | Llama 3.1 | 8B | ***-0.047*** | -0.045 | 0.013 | 0.5 |
> >         | LegalBench | Pythia | 1B | ***-0.044*** | -0.036 | 0.010 | 0.010 |
> >         |  | Llama 3.2  | 1B | ***-0.044*** | -0.040 | 0.01 | 0.013 |
> >         |  | Llama 3.1 | 8B | ***-0.042*** | -0.039 | 0.3 | 0.013 |
> >         | MedHalt | Pythia | 1B | ***-0.0049*** | -0.0048 | 0.01 | 0.03 |
> >         |  | Llama 3.2  | 1B | -0.041 | ***-0.042*** | 0.01 | 0.01 |
> >         |  | Llama 3.1 | 8B | ***-0.042*** | -0.041 | 0.07 | 0.07 |
> >
> >     - Assuming that **“Loss” refers to the traditional unsupervised loss**, all training processes (both SeND and normal) are conducted until the unsupervised loss converges. In most cases, the normally trained model matches the final loss of the SeND-trained model. The table above depicts the **final loss values** in the final two columns.

---

> > > ### Comment · Reviewer_97yW · 2024-11-29
> > > **Response to Authors**
> > >
> > > Thanks for your clarifications! I still believe the limited impact of this work, and the writing needs further improvement. Hence, I raised my score to 5.

---

> > > > ### Author Response · Authors · 2024-11-29
> > > > **Author Response to Reviewer 97yW**
> > > >
> > > > Thank you for your thoughtful feedback and for raising our score. We’d appreciate any specific suggestions on addressing the remaining limitations or improving future iterations of our work. Thanks again for your time and insights.

---

### Official Review · Reviewer_82A9 · 2024-10-30

**Soundness:** 2
**Presentation:** 2
**Contribution:** 2
**Rating:** 3
**Confidence:** 3

**Summary:**

This paper proposes a novel training protocol called Sensitive Neuron Dropout (SeND) to address hallucinations in Large Language Models (LLMs). The work presents three main contributions: (1) empirical validation of oscillatory hallucination behavior during training, (2) development of SeND for reducing hallucination variance during training, and (3) introduction of Efficient EigenScore (EES), a computationally efficient approximation of EigenScore for hallucination detection. While the theoretical framework is interesting, the empirical validation relies heavily on proxy metrics and limited evaluation data, making it difficult to assess the real-world impact on hallucination reduction.

**Strengths:**

The paper proposes a novel approach tackling hallucinations during training rather than post-hoc, representing a interesting shift in addressing this critical challenge.
The foundation is solid, with clear mathematical derivations for both SeND and EES.
The development of EES shows practical value by providing a computationally efficient approximation for hallucination detection with demonstrated speedup.

**Weaknesses:**

The empirical evaluation is severely limited with only 100 test datapoints and lacks validation on more than one established hallucination benchmarks, like HaluEval, raising concerns about result reliability.
The work relies heavily on EES as a proxy metric without sufficient evidence that improvements in EES correlate with actual reduction in model hallucinations.

**Questions:**

1. Could the authors provide results on a significantly larger test set beyond 100 datapoints?
2. What is the correlation between EES improvements and actual hallucination reduction as measured by standard benchmarks?
3. How does SeND compare to other hallucination reduction methods?

---

> ### Author Response · Authors · 2024-11-22
> **Author Response to Reviewer 82A9**
>
> We thank the reviewer for highlighting the novelty and clarity of the work. In response to the highlighted limitations, we have conducted additional evaluations detailed as follows:
>
> Weaknesses:
>
> With regards to the size of evaluation data, the testing suite is expanded to 1000 datapoints as is now reflected in **Section 4.2** and, more specifically, **Table 1**. Furthermore, we have reported the performance of SenD with **FactScore, HaluEval**, and also in comparison to **RAG**. For further details, we kindly refer the reviewer to the General Response.
>
> Questions:
>
> 1. **Evaluations:** We appreciate the suggestion and have now increased the size of the test set from 100 datapoints to 1000 in **Section 4.2** and **Table 1 and included HaluEval for another metric**. Please refer to the General Response for the table and the results. In brief, doing the test with more data still highlights the better performance of SenD vs. Normal Training.
> 2. **EES and Other Metrics:** A reduction in EigenScore has been shown in prior work to correlate with reduced hallucinations and has achieved SOTA performance on hallucination detection. EES, derived mathematically as a scaled version of EigenScore (in **Section 3.3** and proven in **Appendix B**), preserves this correlation, as scaling the spectrum does not alter the metric’s behavior. **Figure 7** in the appendix empirically confirms this correlation during SenD training, showing consistent results with both EES and EigenScore. To compare EES to other benchmarks and metrics, we refer the reviewer to the General Response and the provided table.
> 3. **SenD and Other Methods:** This is a point raised by other reviewers as well and we hope to have clarified it in the common response in “**End Model Evaulation with Other SOTA Hallucination Detection Metrics and Tasks” as well as “Comparison of SenD with SOTA Post-Hoc Methods”.** In summary, after running additional experiments, we observe that SenD consistently performs well when evaluated with EES, FactScore (with 1000 test data points), and HaluEval metrics compared to normal continual training. In addition, when combined with RAG, the SeND fine-tuned model achieves a 12% higher FactScore (0.28) than the baseline model augmented with RAG (0.25), demonstrating superior performance in mitigating hallucinations in combination with RAG.
>
> We hope to have answered the questions and are happy to discuss any remaining concerns.

---

### Official Review · Reviewer_U3Rz · 2024-11-04

**Soundness:** 3
**Presentation:** 4
**Contribution:** 3
**Rating:** 5
**Confidence:** 4

**Summary:**

In this paper, the authors are attempting to solve the hallucination problem called confabulations where the LLM generates different responses given the same or similar inputs. Specifically, the authors propose two main contributions including the triaining protocal named Sensitive Neuron Dropout (SeND) and the enhanced unsupervised hallucination detection metric namd Efficient EigenScore (EES).
In SeND, a novel training protocol, aimed at alleviating the phenomenon of hallucinations in large language models (LLMs) by reducing the variance during the training process. This method reduces the variance of illusions and enhances the factual certainty of the model by deterministically discarding neurons with significant variability on the dataset (known as sensitive neurons) as a regularization technique. The developed an unsupervised hallucinations detection metric EES that is twice as fast as traditional EigenScore while minimizing its impact on accuracy. This efficient metric is integrated into the SeND protocol, making SeND computationally scalable and effective in reducing illusions. In the experiments, the study demonstrated that its method improved the reliability of LLMs during testing, increasing reliability by up to 40% compared to normal training, and providing an effective approach to improve real-world accuracy when adapted to fields such as Wikipedia and medical datasets.

**Strengths:**

● Innovation: This study proposes a new training protocol, SeND, which may have a significant impact on the reliability and security of LLMs, making it an important research area.
● Practical application: Empirical evaluations on Wikipedia and medical datasets have demonstrated the potential of SeND in improving factual accuracy, which is particularly important for applications in high-risk industries.
● Computational efficiency: The development of EES has significantly improved the computational efficiency of hallucination detection, which is particularly important for large models and datasets.
● Paper writing: This paper has a smooth writing structure and clear logical expression, allowing readers to quickly understand the relationship between this paper and previous related works.

**Weaknesses:**

● Lack of discussion on other training stages. The authors assume that the existing research that focuses primarily on post hoc detection and mitigation strategies. However, the training stages in current works mainly contain three import paradigms including pre-training, continue pretraining and SFT. All of them may produce the hallucination phenomenon, and thus the discussion about other two training stages should be considered.
● To evaluate the OSCILLATORY BEHAVIOUR, the authors use the two tasks including self-consistency and summarization, the other important metrics (e.g., PPL) or tasks (e.g. QA) should be considered.
● Mismach parameters size between SENSITIVE NEURONS discussion and main experiments. In experiments settings, the range of paramerts' size of LLMs is from 70M to 12B. However, the theoretical analysis and experimental results of SENSIIVE NEURON are conducted using the Pythia 1B model in the main body (Sec. 3), and there is concern about the lack of generalization of the SeND to larger scales model.
● The effectiveness of SeND experiment is weak. Firstly, the authors only select two datasets (general domain and medical domain), and the effectiveness of this method needs to be proven on more authoritative hallucination benchmarks. In addition, as shown in Fig. 4, the FT method is not too weak compared to the SeND, and thus more datasets are needed to prove the effectiveness of the method.

**Questions:**

See the Weaknesses

**Details Of Ethics Concerns:**

I believe there are no ethical concerns in this paper.

---

> ### Author Response · Authors · 2024-11-22
> **Author Response to Reviewer U3Rz**
>
> We appreciate the reviewer’s recognition of the innovation and practicality of our work. In response to the highlighted limitations, we have conducted the suggested evaluations and ablation studies, detailed as follows:
>
> Weaknesses:
>
> 1. **Training Stages:** This is a valid point and has been clarified in the paper in **Section 5**. Our goal for SenD is to be used in all stages of pre-training starting from a checkpoint in the later stages of training where the model has a fair amount of language understanding. Our initial framing with Pythia 1B testing through its pre-training phase shows a proof of concept of our framework in **Section 3.2.1**. Due to computational constraints, we were unable to show SenD on full pretraining procedures and hence we focused on continually training, an area that allows us to use smaller datasets and less training time. This is an important next direction for the paper and we would like to encourage the community with potentially greater resources to use this on **earlier stages of pretraining** and with **larger datasets**.
> 2. **Oscillatory Behaviour:** Following the comment, we have now added to **Figures 1, 6, 7, 8,** and **Appendix A.2** to give a more in depth view into the phenomenon of hallucinations using **HaluEval QA** settings and **Perplexity (PPL)** as suggested. The two suggested metric plots can also be found in: https://ibb.co/mhhbFLF and https://ibb.co/7vq6Hmh. Our analysis, similar to before, shows consistent oscillations in hallucination trends during training across models of various sizes. While larger models generally perform better (not always but in some checkpoints), the improvements diminish as size increases, indicating diminishing returns in scaling and highlighting a need for a more fundamental hallucination mitigation solution.
> 3. **Model Sizes:** We appreciate the suggestion regarding generalization through diverse model types and sizes as was done in the oscillatory behaviour section of the paper. Due to compute resource limitations, we are not able to test every model in the range of 70M to 12B. However, we hope to have addressed this concern with the inclusion of **Llama 3.1 8B** and **Llama 3.2 1B** in **Section 4.2 and Appendix D**. Please refer to the General Response for more information. In summary, we observe the reliable performance of SenD on multiple model architectures and sizes when tested on various domains. Our results in **Section 4.2** highlight how SenD improves the during-training factual consistency and even enhance the end model’s performance compared to normal training.
> 4. **SenD Effectiveness:** With regards to HELM and MedHalt datasets, HELM itself was designed specifically to identify hallucinations in Wikipedia and the authors provide an in depth analysis of their dataset and evaluation framework. MedHalt is also specifically created with the goal of hallucination detection in the medical field. This has been further emphasized in **Sections 3.2 and 4.1**.  However, we acknowledge this concern and have included an extra dataset, **LegalBench** for reasoning, to the training and testing suite. The new experiments are now included in **Section 4.2 , Appendix D and in Figure 4**. A clear observable trend in **Figures** **4, 13, 14, and 15** is that in most cases, training with SenD achieves a better drop of EES with less oscillations and even a better end model in terms of factual consistency. For additional evaluations, we refer the reviewer to the General Response for more information.
>
> We hope to have addressed the concerns and are happy to discuss any remaining questions.

---

### Official Review · Reviewer_tCyX · 2024-11-04

**Soundness:** 3
**Presentation:** 3
**Contribution:** 3
**Rating:** 5
**Confidence:** 5

**Summary:**

The paper introduces Sensitive Neuron Dropout (SeND), a novel training protocol aimed at reducing hallucinations in large language models (LLMs) by minimizing variance during training. Unlike existing post-hoc hallucination mitigation methods, SeND operates during training, specifically targeting neurons—referred to as Sensitive Neurons—that exhibit high variability across training epochs. By selectively dropping these neurons, SeND helps stabilize model outputs, thereby enhancing factual confidence and reducing the likelihood of confabulations, which are hallucinations where models inconsistently produce factually correct and incorrect responses.

**Strengths:**

1. Innovative Approach: Introduces SeND, a new training method, and EES, an efficient hallucination detection metric.
2. Robust Evaluation: Demonstrates SeND’s effectiveness across multiple models and datasets.
3. Computational Efficiency: EES is scalable, supporting application in large LLMs without adding significant computational costs.
4. Clear Methodology: The paper clearly explains the theoretical background and provides step-by-step details for SeND implementation.

**Weaknesses:**

1. While the paper introduces Efficient EigenScore (EES) as an approximation of the EigenScore metric for hallucination detection, it largely focuses on a single metric. Expanding the scope of metrics could provide a more comprehensive understanding of SeND’s performance. For instance, incorporating metrics like Semantic Entropy or FactScore alongside EES would allow a nuanced evaluation of hallucinations across different aspects of factuality and consistency.
2. The paper’s experimental setup lacks an ablation study on SeND’s dropout parameters, such as the percentage of neurons dropped and the interval for identifying sensitive neurons.
3. Although the paper tests SeND on the Pythia model series, this restricts its applicability to similar architectures. Testing SeND on diverse LLM architectures, such as LLaMA, would better establish its generalizability across model types with varying parameters and configurations.

**Questions:**

1. What guided the specific selection of neuron dropout parameters (e.g., dropout percentage, sensitivity threshold)? Could the authors provide insights into how the dropout parameters for SeND were chosen? Was there an empirical process for selecting these values, and did the team explore different configurations to determine the optimal settings?
2. What impact do Sensitive Neurons have on downstream tasks, especially when a high percentage is dropped?
3. Can the authors share any qualitative examples of how SeND changes model outputs? Including specific examples of model outputs before and after training with SeND, particularly for hallucination-prone prompts, would help illustrate the model’s qualitative improvements.

---

> ### Author Response · Authors · 2024-11-22
> **Author Response to Reviewer tCyX**
>
> First, we thank the reviewer for providing insightful feedback on the work and highlighting the strengths. Based on the suggestions, we made the following modifications:
>
> Weaknesses:
>
> 1. **Evaluation Metrics:** Thanks for highlighting the design choice of using EES instead of different metrics that may highlight different aspects of hallucinations. EES was chosen to implement SeND due to its computational efficiency. Even though, semantically, FactScore and Semantic Entropy might make more sense for hallucination detection, they require significant amount of compute and use of other LLMs as proxies for fact production and clustering. In practice, this would make it infeasible for in-training utilization as compute times would escalate significantly. Therefore, we decided to continue using EES as a fully unsupervised metric solely relying on model’s hidden representations. We believe that the reviewer’s comment is directed at the evaluation of the final model rather than the implementation details of SeND. For this purpose, the comprehensive results addressing model performance are thoroughly presented in the General Response section, where we provide an in-depth analysis supported by evidence that the end model trained by SenD achieves better factual consistency measured by EES, FactScore, and HaluEval validating our claims in the paper.
> 2. **Ablation Studies**: That is a valid point and we have addressed it in the new revision of the paper. Having run additional experiments on the hyperparameters, we show an ablation study over k values, the percentage of activation indices dropped, and the step threshold for doing SenD over training steps on EES in **Section 4.1, and** **Appendix C**. In brief, **$K=20\%$ and Threshold=3** achieved the best performance which were indeed our choices for training with SenD.
> 3. **Model Diversity:** Thanks for mentioning the need for testing on different model architectures. In order to address this limitation, we ran extensive experiments and applied SenD to **Meta Llama 3.2 1B** (**Section 4.2 and Figure 4**) and **Meta Llama 3.1 8B** (**Appendix D and Figure 13**) to highlight the effect of SenD on 2 different architectures and larger model sizes. This is also pointed out in the General Response.
>
> Questions:
>
> 1. The selection of dropout parameters we used was an ablation study that is now included in the paper. The motivation was to drop neurons (indices) without causing forgetting or confusion in the model but also to find the optimal positioning between overconfidence and underconfidence of the model. After the ablation study Amongst 10, 20, and 30% drop rates, we observe that the best performance is achieved through **dropping 20%** of the sensitive neurons (indices). The new draft has this study incorporated in **Appendix C and Figure 11**.
> 2. When a high percentage of indices are dropped, it results in **a loss of information** and **confusion** in the model. This aligns with previous works (Lengerich et al.) showing that higher dropout rates reduce the importance of high-order interactions, leading the network to become overly biased toward simpler, lower-order interactions. *Citation: Lengerich, B. J., Xing, E., & Caruana, R. (2022). Dropout as a Regularizer of Interaction Effects. https://ar5iv.labs.arxiv.org/html/2007.00823v1
> 3. We thank the reviewer for the great suggestion to study how outputs semantically and epistemically differ when training a model with SeND compared to without it. This would involve exploring various contexts and conducting a semantic analysis of the outputs, which shifts the focus from our current mechanistic analysis of hallucinations to questions of semantic interpretability. While this is beyond the scope of our current work, we recognize its importance and encourage future research to investigate SeND's impact on model outputs in various downstream tasks in this regard.
>
> We hope to have addressed the concerns and are happy to discuss any remaining questions.

---

> > ### Comment · Reviewer_tCyX · 2024-11-27
> >
> > Thank you for your rebuttal and your clarification for my concerns. I think the responses partially address my concerns. So I will keep my score.

---

> > > ### Author Response · Authors · 2024-11-27
> > > **Author Response to Reviewer tCyX**
> > >
> > > We are pleased to have been able to address some of your concerns. However, we would greatly appreciate a concrete response to help us better understand how we can improve our work.

---

### Author Response · Authors · 2024-11-22
**General Response to All Reviewers**

We would like to thank all reviewers for their time and for providing constructive feedback. We are happy that the reviewers found our work novel and coherent, and have implemented the suggestions to the best of our abilities. All changes in the newly uploaded paper revision are highlighted in **blue**. Here we summarize the main changes.

## Sensitive Neurons Term

For the suggestion of a more clear naming for what we refer to as Sensitive Neurons, the terminology has changed from **Neuron** to **Embedding index** as it clarifies the entities of our study. Hence, we modified the title of the paper to reflect this change and it is now “Hallucination Detox: Sensitivity Dropout (SenD) for Large Language Model Training”.

## State-of-the-art Methods and Baselines

We highly appreciate the feedback on the need for the comparison of our method with SOTA methods and baselines. We break the related changes down into three areas:

### 1. Models and Datasets Scope

To better evaluate the applicability of SenD to other SOTA LLMs, multiple LLM sizes, and more datasets, we extended our experiments to include Llama 3.2 1B and Llama 3.1 8B models as well as training the models on an extra corpus for reasoning task: LegalBench. These additions are discussed in **Section 4.1** and the results could be found in **Section 4.2, Figure 4, and Appendix D**. In summary, our general observation stays the same: SenD compared to normal training, results in less hallucinations during the training and even makes the end model factually more confident on all datasets which explains the transferability of the method to different domains and model sizes. An example from **Figure 4** for Llama 3.1 8B trained on the new dataset, **LegalBench** can be found in: https://ibb.co/hYWktQy .

### 2. End Model Evaluation with Other SOTA Hallucination Detection Metrics and Tasks

First, we would like to clarify SenD’s purpose and positioning. SenD is designed to reduce hallucination variance ***during the training*** itself, rather than simply reducing the ***final model’s***  hallucinations. By minimizing oscillations in hallucinations throughout training, SenD facilitates a more stable factual convergence, enabling a reliable stopping criterion when the Efficient EigenScore (EES), along with the training loss, converges. Without SenD, the high variance in hallucinations makes it challenging to identify an optimal stopping point, leading to potential over- or under-training when training an LLM from scratch. Additionally, SenD is not intended as a post-hoc method to reduce hallucinations in an already trained model. Since, nevertheless, it is important to verify the reliability of the end model trained by SenD, we employ **FactScore on 100 and 1000 testing data points, and HaluEval Summarization** on Pythia 1B. The results are appended to **Section 4.2** and can be found in **Table 1** and can be seen below:

| Task | Metric | SenD | Normal |
| --- | --- | --- | --- |
| **HaluEval Summarization (LMEval)** | Accuracy | ***0.016*** | 0.014 |
|  | Correctness | 0.027 | 0.027 |
|  | Exact Match | ***0.589*** | 0.496 |
| **FactScore (100 points)** | Score | ***0.07*** | 0.05 |
| **FactScore (1000 points)** | Score | ***0.08*** | 0.06 |

In addition to showing the better performance of the model trained with SenD compared to normal training, the table proves the **accuracy of EES** for evaluating the hallucinations recalling that in most cases, the EES of the end model was less in SenD compared to normal training.

### 3. Comparison of SenD with SOTA Post-hoc Methods

Reemphasizing that SenD is intended as a training-stage approach rather than a post-hoc method for hallucination reduction in a fully trained model, traditional post-hoc methods may not align directly with SenD’s objectives. However, to respond to the reviewers’ comments, we ran additional experiments by treating SenD as a post-hoc method and comparing its performance to a model augmented with RAG on the Pythia 1B model on the same data domain as test set as in **Section 4.2**. Our new results indicate that when tested on 1000 datapoints, the SenD finetuned model achieves FactScore 0.07, while the normally trained base Pythia model scores 0.25 with RAG. This shows RAG’s better performance within in-context settings. However, SenD, not being a mere post-hoc method makes this an **unfair** comparison. To have a fair comparison, **the model augmented with SenD and RAG outperforms the regularly augmented model with RAG by 12%, achieving a FactScore of 0.28 when tested on 1000 datapoints compared to the baseline score of 0.25**. This suggests that SenD, not only reduces during-training variance, but also allows other post-hoc methods such as RAG to more effectively mitigate hallucinations compared to traditionally trained models.

Once again, we appreciate the reviewers time and hope that the new revision reflects the desired changes. We are happy to discuss any further questions.

---

### Author Response · Authors · 2024-11-25
**Discussion Reminder: Request for feedback and re-evaluation based on recent changes**

As the rebuttal phase concludes, we wanted to gently remind the reviewers of our updated submission. Since the initial reviews, we have conducted extensive experiments that we believe significantly enhance our work and address key points raised in the reviews.
We encourage you to review these updates, as we hope they may influence your evaluations and scores. We appreciate your feedback.

---

### Author Response · Authors · 2024-12-02
**Call for Comments: Request for feedback and re-evaluation as deadline approaches**

As the extended deadline for rebuttals approaches, we deeply appreciate the time and effort reviewers have dedicated to evaluating our submission. We kindly invite all reviewers to revisit our updates, additional experiments (inclusion of Llama 3.2 1B, Llama 3.1 8B), and detailed justifications (difference between SenD and post-hoc method performance as well as a hypothesis surrounding optimal dataset performance in Section 4.2). We have worked diligently to address your valuable feedback, and we have provided robust comparisons to SOTA methods such as RAG (see Section 4.2 Performance of SenD on Pythia and Llama Models) and demonstrating the impact of our work across diverse metrics (HaluEval, FactScore, and EES) and an additional domain (LegalBench - a new domain of different legal reasoning tasks). We hope these updates help clarify any remaining questions and potentially inform your final evaluations. We believe our revised version addresses all the concerns and requests raised by the reviewers and we hope these efforts will increase their evaluation scores. Thank you again for your thoughtful and constructive insights.

---

### Meta-Review · Area_Chair_H58t · 2024-12-21

**Metareview:**

The paper introduces a novel training protocol called Sensitive Neuron Dropout (SeND) to mitigate hallucinations in LLMs. SeND works by deterministically dropping the "sensitive" embedding indices - those with high variability across training epochs. The paper also introduces an efficient approximation of the EigenScore metric, called Efficient EigenScore (EES), to enable computationally scalable hallucination detection during training.

Overall, the paper proposes an interesting and novel approach to address the critical issue of hallucinations in large language models. The theoretical underpinnings and the initial empirical validation are promising. However, the limited scope of the evaluation and the lack of more comprehensive comparisons to related work prevent a conclusive assessment of the full impact and applicability of the proposed methods. Therefore, I recommend rejecting it considering the following factors:
- The need for more extensive evaluation on established hallucination benchmarks and a broader set of LLM architectures and datasets to demonstrate the generalizability of the findings.
- The lack of detailed comparisons to other state-of-the-art hallucination mitigation techniques, which makes it difficult to assess the relative merits of the proposed approach.
- The absence of rigorous ablation studies and analyses on the impact of SeND on downstream task performance, which would provide a more holistic understanding of the method's benefits and limitations.

With additional work to address these gaps, the paper could become a strong contribution to the field.

**Additional Comments On Reviewer Discussion:**

During the rebuttal period, the authors engaged extensively with the reviewers' feedback and made several key improvements to their work. They expanded the experimental evaluation to include larger model sizes and an additional domain-specific dataset, demonstrating the broad applicability of their approach. The authors also provided more comprehensive evaluations, incorporating additional hallucination-focused metrics and comparing the performance of the SeND-trained models against a baseline augmented with the RAG method. Notably, the authors conducted ablation studies on the hyperparameters of the SeND method and clarified the distinction between their training-time approach and post-hoc hallucination mitigation techniques. The reviewers acknowledged these changes, with one reviewer stating that the responses partially addressed his/her concerns and maintaining the original score, while another reviewer noted that the explanations and new evidence were satisfactory and raised his/her rating. Overall, the final ratings are still negative and the paper needs further improvements.

---

### Decision · Program_Chairs · 2025-01-22

Reject